# A Three-Dimensional Structured Light Vision System by Using a Combination of Single-Line and Three-Line Lasers

**DOI:** 10.3390/s23010013

**Published:** 2022-12-20

**Authors:** Qiucheng Sun, Zeming Ren, Jinlong Zhu, Weiyu Dai, Mingze Wang, Mingyu Sun

**Affiliations:** College of Computer Science and Technology, Changchun Normal University, Changchun 130032, China

**Keywords:** multi-line structured light, three-dimensional measurement, visual measurement

## Abstract

A multi-line structured light measurement method that combines a single-line and a three-line laser, in which precision sliding rails and displacement measurement equipment are not required, is proposed in this paper. During the measurement, the single-line structured light projects onto the surface of an object and the three-line structured light remains fixed. The single-line laser is moved and intersects with the three-line laser to form three intersection points. The single-line light plane can be solved using the camera coordinates of three intersection points, thus completing the real-time calibration of the scanned light plane. The single-line laser can be scanned at any angle to determine the overall complete three-dimensional (3D) shape of the object during the process. Experimental results show that this method overcomes the difficulty of obtaining information about certain angles and locations and can effectively recover the 3D shape of the object. The measurement system’s repetition error is under 0.16 mm, which is sufficient to measure the 3D shapes of complicated workpieces.

## 1. Introduction

At present, 3D measurement is widely used in the fields of heritage restoration and machinery production [1]. Among them, 3D object surface contour measurement methods are generally divided into two types, contact and non-contact. The current contact measurement methods, such as coordinate measuring machines [2], can basically meet the requirements of measurement accuracy, but the contact measurement methods have problems such as slow measurement speed and easy damage to the contact surface, which limit their measurement accuracy. Non-contact measurements comprise projected structured light, interferometry, ultrasonic measurement, and other techniques for applications in the field of quick measurement [3,4,5].Structured light projection 3D measurement technology, for example, is easy to use and may provide accurate real-time measurements. In structured light projection 3D measurement technology, laser projection 3D measurement technology is a main research technique. This technique is gradually displacing contact measuring methods in a variety of industries, including car assembly [6], reverse engineering, and part quality inspection [7,8,9].

Currently, line laser is mainly used in laser projection 3D measurements to scan and reconstruct the object under test [10,11,12]. The line structured light system emits a laser beam that forms a strip projection on the surface of the object. Afterwards, the camera captures the laser stripe and extracts the center of the laser stripe. Finally, the system converts the light strip pixel information into physical information based on the laser plane equation and camera parameters and fits these 3D point cloud to obtain the 3D shape of the object [13,14,15,16].

To calibrate the structured light planes, Dewar [17] projected the laser onto some fine non-coplanar threads mounted in 3D space, and then defined the structured light planes by obtaining images of bright light points from the camera as control points. Huang [18] presented a new calibration method for line structured light multi-vision sensors based on combined targets. Each line structured light multi-vision sensor is calibrated locally with high accuracy by combining targets. The position of the target has no impact on the profile feature imaging for ball targets. Using a single ball target as his foundation, Zhou [19] proposed a technique for the quick calibration of a line-structured light system. Xie [20] used a planar target and a raising block to complete the field calibration of the linear structured light sensor. The three-dimensional measurement method based on line-structured light can only measure 3D contour information on one cross-section of an object at a time. Therefore, motion devices are required to sweep the laser plane across the surface of the measured object to gather 3D data on the complete surface [21]. Both translational and rotational scanning methods used slides and rotary tables and are commonly used to create these 3D measurement systems.

To estimate the size of big forgings in thermal condition, Zhang [22] used high-precision guides in combination with line structured light. The experimental results revealed that the procedure is effective with an error of less than 1 mm, which meets the accuracy criteria. For remote inspection of internal delamination in wind turbine blades, Hwang [23] presented a continuous line laser scanning thermography (CLLST) method. The test results have shown that the 10 mm diameter internal delamination located 1 mm underneath the blade surface was successfully detected even 10 m from the target blade. The above methods usually require the scanning direction to be strictly perpendicular to the structured light plane, in which the strict position relationship is too difficult to achieve in practical measurements. Then, a number of improved methods have been proposed. Wu [24] designed and manufactured a calibration device through the perspective invariance and the projective invariant cross-ratio properties. Experimental results with different test pieces demonstrated that the minimum geometric accuracy was 18 μm and the repeatability was less than ±3 μm. Zeng [25] used a planar target to calibrate the translation vector, which solved the traditional inconvenience of calibrating the precise movement direction by high-precision auxiliary equipment. Lin [26] proposed a binocular stereo vision based on the camera translation direction cosine calibration method to realize the conversion of point clouds from local to a global coordinate system, so as to obtain the 3D data of the object under test. When it is necessary to measure the inner surface of an object or complete information about the outside of an object, a series of methods were proposed in which rotary tables or rotating guides were applied. Cai [27] used a line laser in combination with a motor, measuring the 3D shape of potatoes, which can help farmers to analyze their phenotypic characteristics and grade them. Guo [28] used a high precision air floating rotary table and a line structure light sensor to rapidly obtain information on the 3D shape of gear. Liu [29] placed a mouse on a metal part on a rotating platform with a motor to measure its 3D shape by modulated the average stripe width within a favorable range. Measurements on different parts show that the method improves the integrity of the surface. Wang [30] put the pipe on a turntable and the linear structured light measuring device on the middle of the pipe. The three-dimensional measurement of the inner wall of the pipe was realized by rotating the pipe. In panoramic 3D shape measurement based on a turntable, the point cloud registration accuracy will be influenced to some extent by the calibration of the rotation axis. Therefore, some methods of calibrating the rotation axis have been proposed. Cai [31] proposed an auxiliary camera-based calibration method. The experimental results show that the method improves the calibration accuracy of the rotation axis vector. Although all of the approaches above provide reasonable measurement precision, they are extremely dependent on the equipment’s accuracy, both in the experimental setup and during the scanning process. For example, the slide’s sliding accuracy, the rotary stage’s rotation accuracy, and the stepper motor’s displacement accuracy will all have an impact on the final measurement results. Furthermore, in these methods, the angle of light plane scans is frequently fixed and cannot be adjusted during the scanning process, which can be easily affected by noise and folds, resulting in missing image information in certain areas.

In order to lessen the inaccuracies caused by the equipment, several academics have proposed using handheld laser measurement systems. Simon et al. [32] presented a low-cost system for 3D data acquisition and fast pairwise surface registration. The object can simply be placed under a background plate with a priori information, and the laser can be held in the hand to obtain the complete 3D shape of the object. However, this method often requires a background plate larger than the object when measuring the object, which is not conducive to industrial field measurements.

This paper completes the construction of a multi-line structured light 3D measurement system combining single-line and three-line lasers. The procedure can be summarized as: first, a structured light system is calibrated, which includes a monocular camera and a three-line structured light. Second, the pixel coordinates of three intersection points are detected, which are generated when single-line structure light intersects with three-line structure light during scanning. Finally, the single-line light plane can be calibrated in real-time by means of the camera coordinates of the three intersection points, which are solved by the three-line structured light system. During the measurement, the single-line structured light can be scanning at any angle, ensuring that no information is lost in a particular location of the object. The difficulty of obtaining information about certain angles and locations when the laser travels on a set track is overcome, and the demand for equipment, especially precision mobile equipment, is reduced.

The organization of this paper is as follows. Section 2 describes the principle of multi-line structured light measurement system. Section 3 introduces the calibration of structured light systems. Section 4 describes the method of 3D measurement. Finally, the experimental results and conclusions are presented in Section 5 and Section 6, respectively.

## 2. The Principle of Multi-Line Structured Light Measurement System

The measurement system designed in this paper consists of three main components: a laser emission module, an image acquisition module and an image processing module, as shown in Figure 1. The laser transmitter module consists of a three-line laser and a single-line laser. The image acquisition module captures line laser streak images that are distorted due to height fluctuations on the surface of the object, mainly through a charge-coupled device (CCD) image sensor. The image processing module performs the subsequent processing of the image acquired by the CCD camera, for example, by obtaining the pixel coordinate values of each light stripe center in the image. As a result of the spatial relationship between the world coordinate system, the camera coordinate system, and the image coordinate system, in combination with the pixel coordinate system, the coordinates of the object in the camera coordinate system can be obtained by a coordinate transformation for non-contact measurement. The coordinate relationships are shown in Figure 2.

As shown in Figure 3, the measurement scheme of the system consists of two steps: system calibration and 3D measurement. The system calibration includes camera calibration and three-line light plane calibration. The camera is calibrated according to the acquired calibration plate image to obtain the internal parameters of the camera. Next, the calibration of the three-line light plane is completed by a flat target with a black square pattern. Then, the equation of the single-line light plane can be derived from the three intersections of the three-line light plane and the single-line light plane. During the 3D measurement, the centerline of the light-strip image of the object surface can be extracted using the Steger algorithm [33]. Then, the extracted pixel coordinates are converted to 3D coordinates in the camera coordinate system by combining the camera internal parameters and the equation of the single-line light plane to complete the 3D measurement of the object.

## 3. Calibration of Structured Light Systems

The calibration of the structured light system in this paper is divided into two parts. The camera calibration is the first step to obtaining the camera internal parameters, and the three-line light plane calibration is the second step to obtaining the three-line light plane equations.

### 3.1. Camera Calibration

Figure 4 shows a schematic diagram of a typical coordinate system. P is a random point in 3D space, the world coordinates of the change point are ($X_{w},Y_{w},Z_{w}$), the camera coordinates are ($X_{c},Y_{c},Z_{c}$), Pd is the actual projection of P on the image, and Pu is the ideal projection point. The image coordinates are $\left(x_{d},x_{d}\right),\left(x_{u},x_{u}\right)$, and (u,v) are the pixel coordinates of P. The positional relationship [34] between the world coordinate system and the camera coordinate system can be denoted by
(1)$$Z_{c}\left[\begin{array}{c} u\\ v\\ 1 \end{array}\right]=\left[\begin{array}{cccc} \alpha & \gamma & u_{0} & 0\\ 0 & \beta & v_{0} & 0\\ 0 & 0 & 1 & 0 \end{array}\right]\left[\begin{array}{cc} R & t\\ 0^{T} & 1 \end{array}\right]\left[\begin{array}{c} X_{w}\\ Y_{w}\\ 0\\ 1 \end{array}\right]=\left[\begin{array}{ccc} \alpha & r & u_{0}\\ 0 & \beta & v_{0}\\ 0 & 0 & 1 \end{array}\right]\left[r_{1}\;r_{2}\;t\right]\left[\begin{array}{c} X_{w}\\ Y_{w}\\ 1 \end{array}\right]$$

In order to obtain more accurate camera parameters, a non-linear imaging model of the camera is used [35]. Taking into account the effect of radial and tangential aberrations, the meaning of the actual projection coordinates is as follows.
(2)$$\left\{\begin{array}{c} \delta_{x}\left(x_{u},y_{u}\right)=x_{u}\left(k_{1}p^{2}+k_{2}p^{4}\right)+2p_{1}x_{4}y_{u}+p_{2}\left(3x_{u}^{2}+y_{u}^{2}\right)\\ \delta y\left(x_{u},y_{u}\right)=y_{u}\left(k_{1}p^{2}+k_{2}p^{4}\right)+2p_{1}x_{4}y_{u}+p_{2}\left(3x_{u}^{2}+y_{u}^{2}\right) \end{array}\right.$$
where $p=\sqrt{\left(x_{u}^{2}+y_{u}^{2}\right)}$, $k_{1},k_{2},p_{1},p_{2}$ are the radial and tangential distortion coefficients, respectively. This produces a nonlinear optimization function. The function can be solved precisely by means of the Levenberg–Marquardt (L–M) algorithm. Finally, the intrinsic matrix $A=\left[\begin{array}{ccc} \alpha & r & u_{0}\\ 0 & \beta & v_{0}\\ 0 & 0 & 1 \end{array}\right]$ and the distortion coefficient $K={\left(k_{1},k_{2},p_{1},p_{2}\right)}^{T}$ are obtained.

### 3.2. The Principle of Three-Line Light Plane Calibration

The three-line structured light’s calibration precision will have a direct impact on the measurement accuracy because it is utilized to position the scanning light plane in real-time.

In this paper, an intersection style multi-line structured light system is mainly used, where the laser throws three light planes intersecting on the same line, and the angle between the light planes is a constant α, as shown in Figure 5.

In order to obtain the light plane equations, this section consists of the following three main steps: the first step determines the external parameters of the flat target with a black square pattern, as shown in Figure 6. The second step uses the Steger method to obtain the coordinates of the center point of the light stripe on the target image. The center point coordinates are the projected into the camera coordinate system by the internal parameters of the camera and the external parameters of the target as shown in Figure 7. The third step uses these three sets of points as control points to determine the equations of the structured light plane with the a priori position relationship of the structured light plane as the constraint which is the angle between the light planes (the constant α = 5°) as shown in Figure 8. The light plane equations are as follows
(3)$$\left\{\begin{array}{c} A_{1}X+B_{1}Y+C_{1}Z+D_{1}=0\\ A_{2}X+B_{2}Y+C_{2}Z+D_{2}=0\\ (A_{1}+\mu A_{2})X+(B_{1}+\mu B_{2})Y+(C_{1}+\mu C_{2})Z+\left(D_{1}+\mu D_{2}\right)=0 \end{array}\right.$$
where A,B,C,D are the light plane equation coefficient, μ is a scale factor.

A uniform objective function is established:(4)$$\begin{array}{l} \Delta =\overset{n_{1}}{\underset{i=1}{\sum }}{\left(A_{1}x_{1i}+B_{1}y_{1i}+C_{1}z_{1i}+D_{1}\right)}^{2}+\overset{n_{2}}{\underset{i=1}{\sum }}{\left(A_{2}x_{2i}+B_{2}y_{2i}+C_{2}z_{2i}+D_{2}\right)}^{2}+\\ \overset{n_{3}}{\underset{i=1}{\sum }}{\left(\left(A_{1}+\mu A_{2}\right)x_{3i}+\left(B_{1}+\mu B_{2}\right)y_{3i}+\left(C_{1}+\mu C_{2}\right)z_{3i}+\left(D_{1}+\mu D_{2}\right)\right)}^{2} \end{array}$$
under the constraint
(5)$$\left\{\begin{array}{c} \frac{\left(A_{1},B_{1},C_{1}\right){\left(A_{2},B_{2},C_{2}\right)}^{T}}{\sqrt{A_{1}^{2}+B_{1}^{2}+C_{1}^{2}}\sqrt{A_{2}^{2}+B_{2}^{2}+C_{2}^{2}}}=\cos2\alpha \\ \frac{{\left(A_{1},B_{1},C_{1}\right)\left(A_{1}+\mu A_{2},B_{1}+\mu B_{2},C_{1}+\mu C_{2}\right)}^{T}}{\sqrt{A_{1}^{2}+B_{1}^{2}+C_{1}^{2}}\sqrt{{\left(A_{1}+\mu A_{2}\right)}^{2}+{\left(B_{1}+\mu B_{2}\right)}^{2}+{\left(C_{1}+\mu C_{2}\right)}^{2}}}=\cos\alpha \\ \frac{{\left(A_{2},B_{2},C_{2}\right)\left(A_{1}+\mu A_{2},B_{1}+\mu B_{2},C_{1}+\mu C_{2}\right)}^{T}}{\sqrt{A_{2}^{2}+B_{2}^{2}+C_{2}^{2}}\sqrt{{\left(A_{1}+\mu A_{2}\right)}^{2}+{\left(B_{1}+\mu B_{2}\right)}^{2}+{\left(C_{1}+\mu C_{2}\right)}^{2}}}=\cos\alpha \end{array}\right.$$
which ensures that the angle between the light planes is constant. Here, $\left(x_{1i},y_{1i},z_{1i}\right)i=1,\ldots ,n_{1},\left(x_{2i},y_{2i},z_{2i}\right)i=1,\ldots ,n_{2}$,$\left(x_{3i},y_{3i},z_{3i}\right)i=1,\ldots ,n_{3}$ represents three sets of 3D points corresponding to three planes. Finally, the coefficients of Equation (6) are solved using the L–M algorithm. In this way, the a priori geometric relation of the planes can be accurately transplanted into the camera coordinate frame.

## 4. Method of Three-Dimensional Measurement

The measurement process in this paper is achieved by single-line structured light scanning. Therefore, real-time calibration of the single-line light plane is essential to the measurement process. The real-time calibration method is divided into three main steps: Step 1: Detect the pixel coordinates of the intersection points

Since the single-line laser must remain intersected by the three-line laser during the measurement, three intersection points are generated. As shown in Figure 9, the centerline of the light stripe in the area near the intersection point is detected, and the coordinates of the intersection point are calculated by fitting the lines L1 and L2.Step 2: Calculate the intersection points in camera coordinates

The three intersection points are located both in the single-line structured light plane and in each plane of the three-line structured light respectively. Therefore, by using the internal parameters of the camera and the three-line structured light plane equation, the three-dimensional coordinates of the three intersection points in the camera coordinate system can be solved.Step 3: Fitting the space plane equation

The space plane equation, which is the single-line light plane equation, can be fitted using the three-dimensional coordinates of the three intersection points, as illustrated in Figure 10.

After completing the real-time calibration, the feature coordinates in a specific plane in the 3D space are obtained by the single-line structured light in the case of a single frame. Multiple frames of data can be registered under the same coordinate system to obtain the original point cloud data of the object to be reconstructed with the single-line laser scanning at any angle.

## 5. Experiment

In this paper, the need for precision sliding rails and other auxiliary equipment in the traditional method is eliminated, and the 3D information of the object can be obtained by simply irradiating two lasers onto the surface of the object to be measured. The measurement system is shown in Figure 11. The configuration used in this study is extremely simple. It generally constitutes a three-line laser (Lasiris; Coherent, Inc., Santa Clara, CA, USA) emitting a Gaussian beam at a wave-length of 660 nm with an output power of 20 mW, a single-line laser emitting a Gaussian beam at a wave-length of 650 nm with an output power of 50 mW (Lei xin Yanxian-LXL65050) and an 8 mm lens camera (MER-1070-14U3M; DaHeng, Beijing, China). The single-line laser can be moved freely during the measurement, and can even be moved by hand. The experiments in this paper consist of camera calibration, three-line structured light calibration, and real-time measurement.

### 5.1. Calibration of the Camera

As shown in Figure 12, the checkerboard target is used, and the size of the checker is 4 × 4 mm. Four sets of calibration experiments were carried out, with nine images acquired by each camera set. The overall mean pixel error of the calibrated images was selected as the basis for judging the camera calibration results.

Finally, a set of camera internal parameters and distortion parameters were obtained.
$$\begin{array}{c} A=\left[\begin{array}{ccc} 4990.2635 & 0.77900164 & 1929.2399\\ 0 & 4993.9024 & 1279.1607\\ 0 & 0 & 1 \end{array}\right],\\ K=\left[\begin{array}{c} 0.0815\\ -0.3809\\ 0.0042\\ -0.0004 \end{array}\right] \end{array}$$

### 5.2. Calibration of Three-Line Light Plane

Nine patterns of the planar target at different orientations in space are acquired by the camera, and three-line laser is needed to be projected on the target synchronously, as shown in Figure 13. Using the method proposed in Section 3.2, the three structured light planes can be solved, and the equations of the planes are as follows:(6)$$\left\{\begin{array}{c} \;0.922x-7.397y-1.429z-1000=0\\ \;1.321x-9.319y-0.992z-1000=0\\ \;1.986x-12.52y-0.263z-1000=0 \end{array}\right.$$

### 5.3. Object Measurement in Real Time

In this experiment, a face mask model and a standard gauge block were measured separately.

#### 5.3.1. Face Mask Model Measurement in Real Time

For comparison purposes, the point cloud of the face mask model obtained by traditional methods using slide rails is shown in Figure 14. It can be observed that the point cloud of face model has relatively few data points within the ear region. This is due to the presence of folds in the ear area, causing occlusions or discontinuities in the scanned image. Unlike traditional methods, the method proposed in this paper allows free control of the scanning direction, thus reducing the influence of the folded area on the image and increasing the effective data information in this area. The point cloud data obtained using the method is shown in Figure 15. The 3D reconstruction images obtained by the traditional method and the method proposed in this paper are shown in Figure 16. The results show that the measurements using the method proposed in this paper are effective.

#### 5.3.2. Standard Gauge Block Measurement in Real Time

In order to determine the spatial measurement accuracy of the system, the example of four measuring blocks with known thicknesses of 70 mm, 60 mm, 50 mm and 40 mm, respectively, is used. The accuracy of a standard part is 0.0015 mm. The real-time scan image is shown in Figure 17. The acquired point cloud image is shown in Figure 18. The point cloud data from the surface of the Gauge block and the point cloud scanned at the bottom are used to fit two parallel planes, and the distance between the planes is the thickness of the gauge block. The fitted spatial plane images are shown in Figure 19. The measurement results are shown in Table 1.

The RMSE is the root mean square error. The STD is the standard deviation. Table 2 shows that between the measurement ranges between 40 mm and 70 mm, the standard deviation is less than 0.11 mm, and the root mean square error is less 0.11 mm, demonstrating the measurement system’s high level of stability. The maximum measurement error for the standard gauge block does not exceed 0.16 mm. In conclusion, it can be stated that the system’s spatial measurement accuracy is 0.16 mm when considering the maximum measurement variation.

## 6. Conclusions

This paper proposes a three-dimensional measurement system that combines a single-line and a three-line laser. The measurement system experimental platform was developed without the need of typical equipment for this technique, such as slide rails and step motors. This method therefore reduces experimental error caused by sophisticated equipment or complicated protocols. Thus, the proposed method is more convenient and straightforward. In addition, when measuring the object from multiple angles, the traditional methods could not deal with folded area (such as a human ear region) well. This issue is solved by the method proposed in this paper. Experiments show that the measurement approach given in this paper can successfully reconstruct the human face mask model and gauge block, as well as highlighting this method’s simplicity and effectiveness. Experiments from measuring blocks of known thickness show that the spatial measurement accuracy of the system is 0.16 mm when the maximum measurement variation is considered.

## Figures and Tables

**Figure 1 sensors-23-00013-f001:**
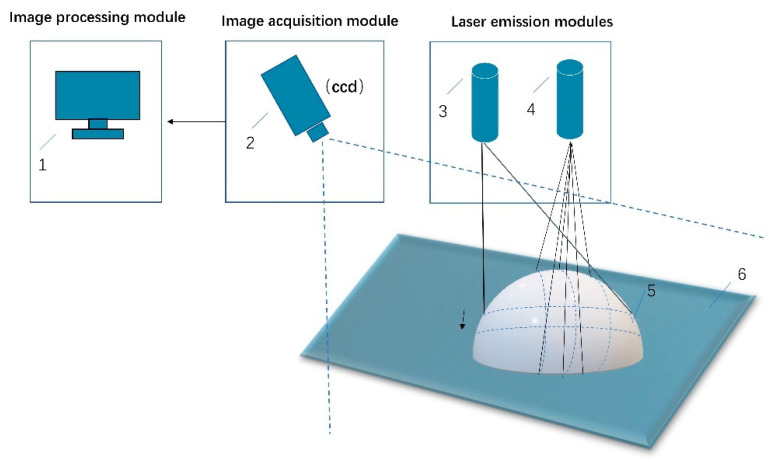
The diagram of the measurement system. (1) Computer, (2) Camera, (3) Single-line laser, (4) Three-line laser, (5) Object, (6) Experiment platform.

**Figure 2 sensors-23-00013-f002:**
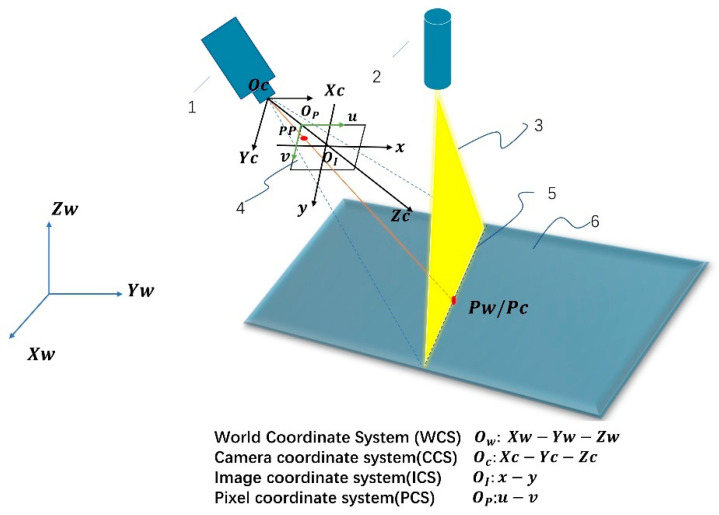
The geometric structure of the line structured vision sensor. (1) Camera, (2) Single-line laser, (3) Laser projector, (4) Image plane, (5) Light stripe, (6) Experiment platform.

**Figure 3 sensors-23-00013-f003:**
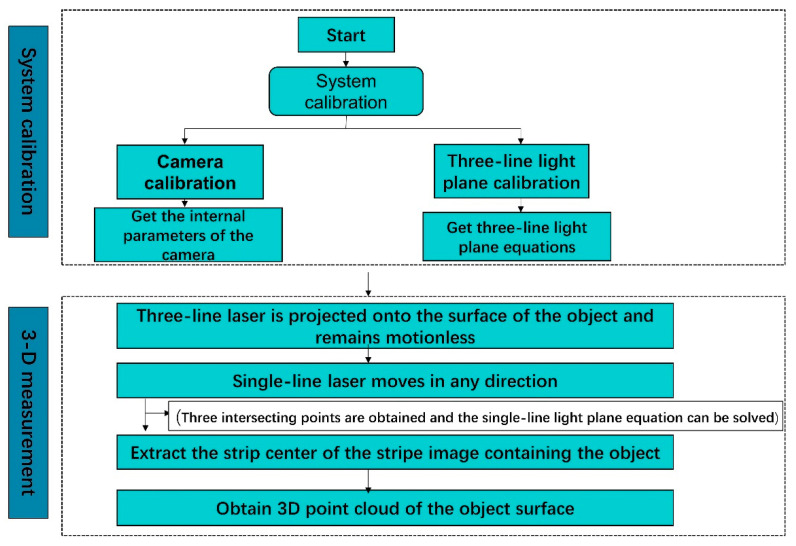
System measurement flow chart.

**Figure 4 sensors-23-00013-f004:**
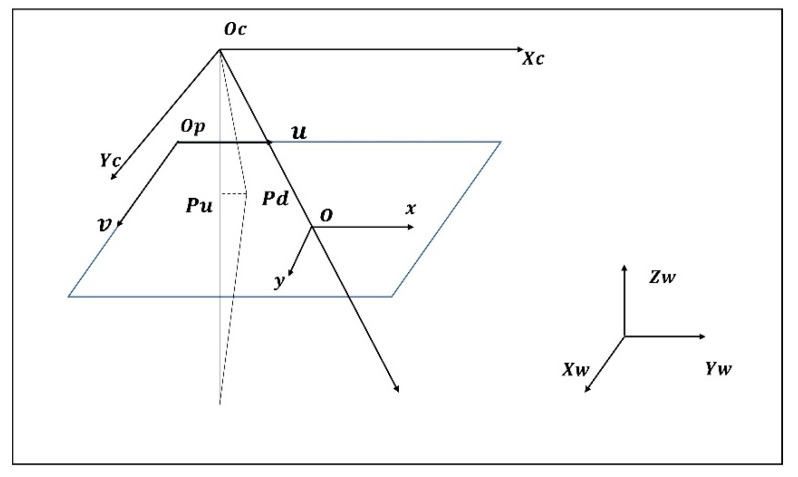
Coordinate system of measurement system.

**Figure 5 sensors-23-00013-f005:**
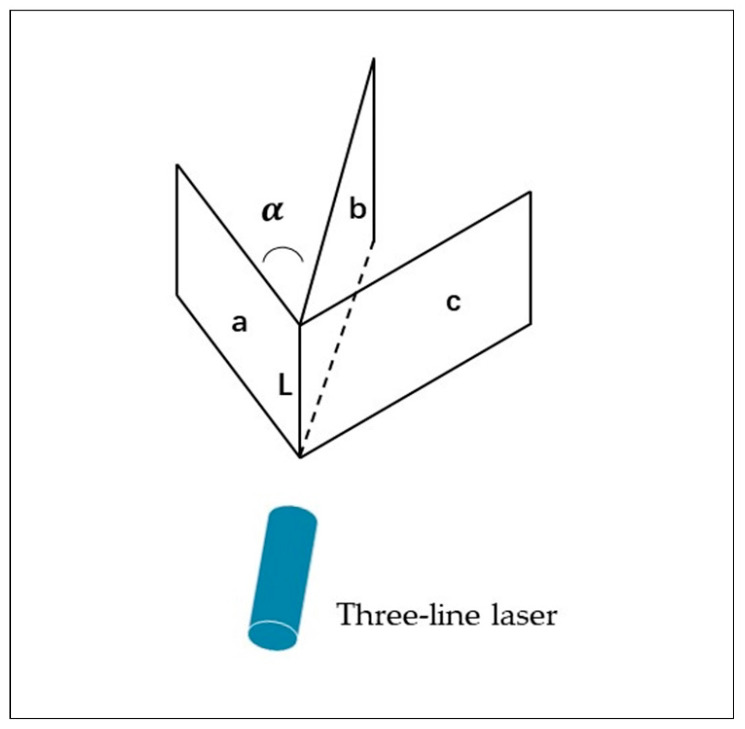
Spatial geometric relationship of the structured light planes.

**Figure 6 sensors-23-00013-f006:**
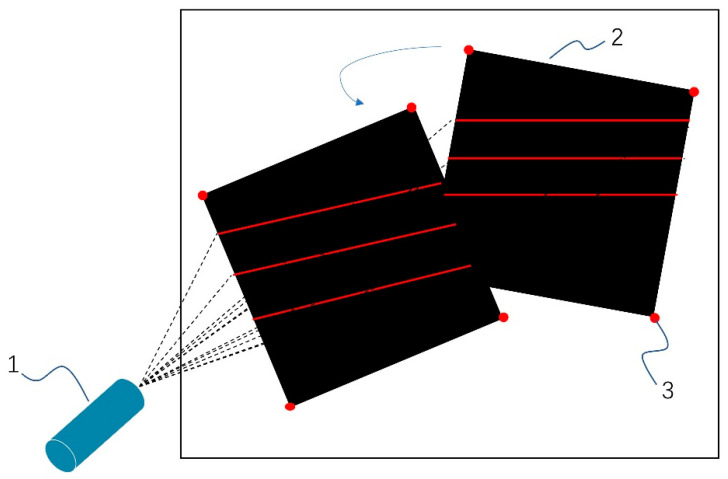
Three-line light plane calibration. (1) Three-line laser, (2) A flat target with black square pattern, (3) Corner Point.

**Figure 7 sensors-23-00013-f007:**
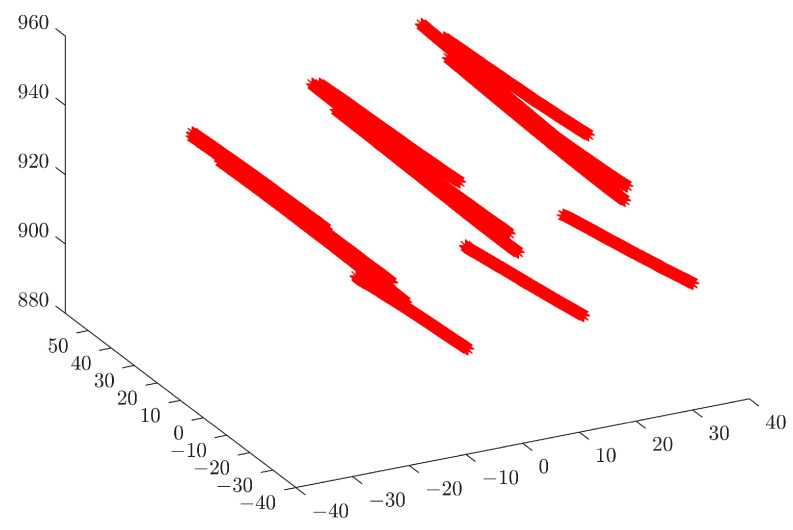
Three sets of 3-D points.

**Figure 8 sensors-23-00013-f008:**
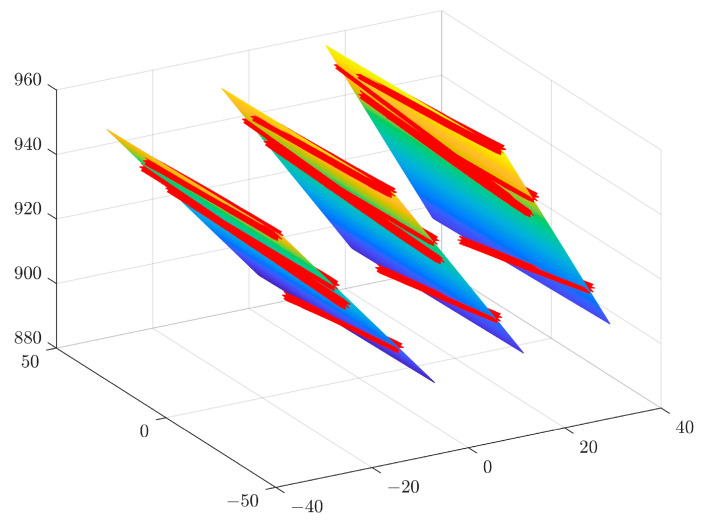
Three light planes in space.

**Figure 9 sensors-23-00013-f009:**
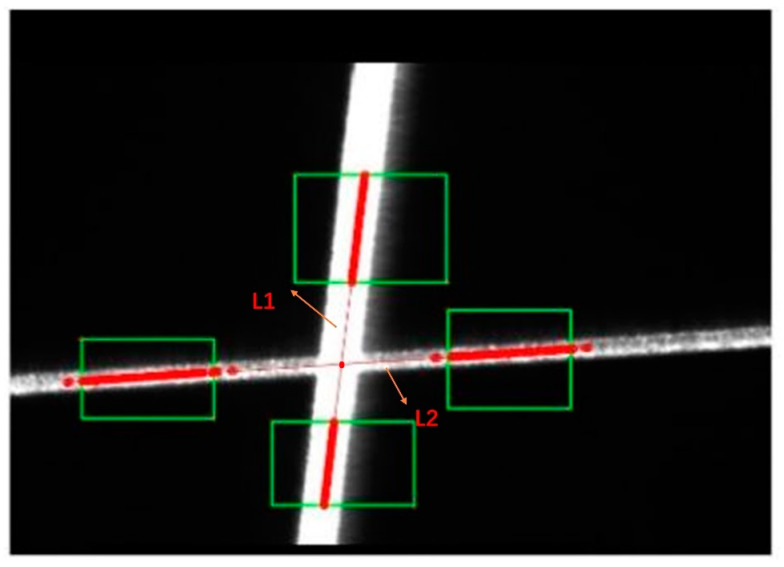
Detect the pixel coordinates of the intersection point.

**Figure 10 sensors-23-00013-f010:**
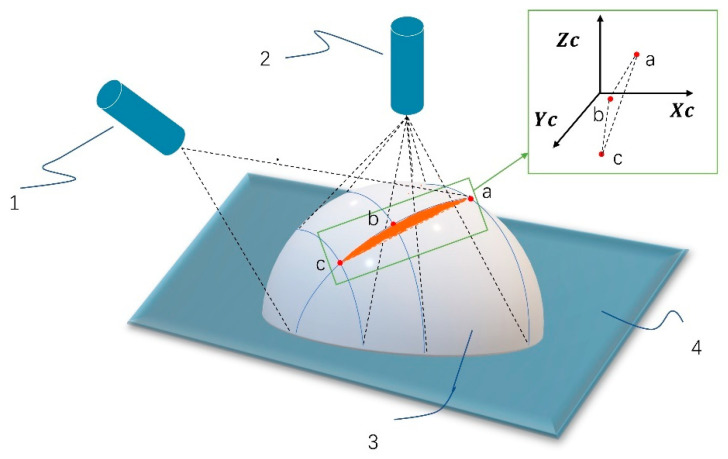
Three points in space determine a plane. (1) Single-line laser, (2) Three-line laser, (3) Object, (4) Experiment platform.

**Figure 11 sensors-23-00013-f011:**
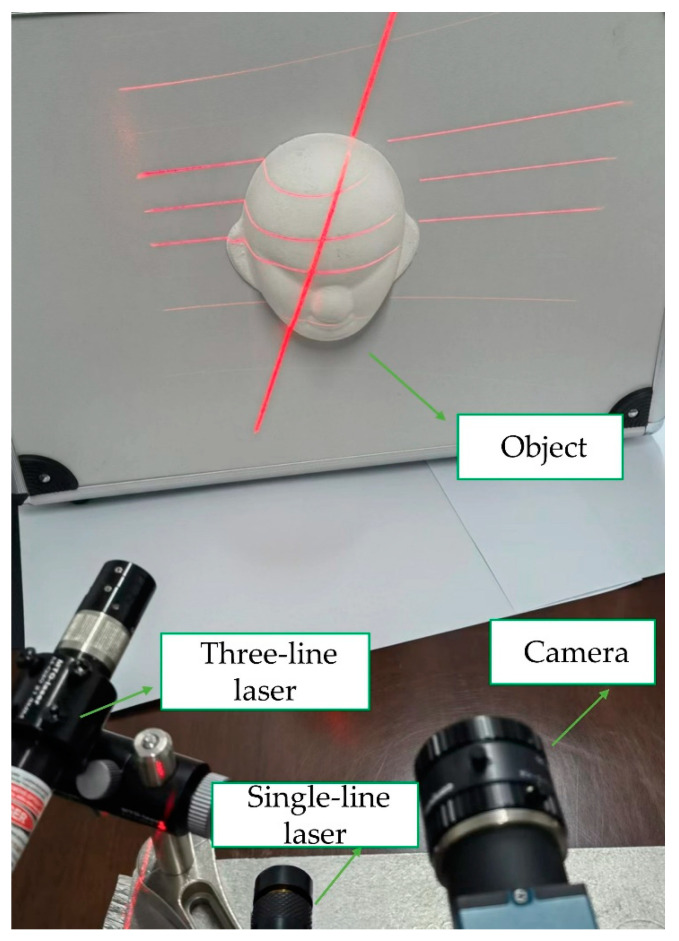
Measurement image of the face model.

**Figure 12 sensors-23-00013-f012:**
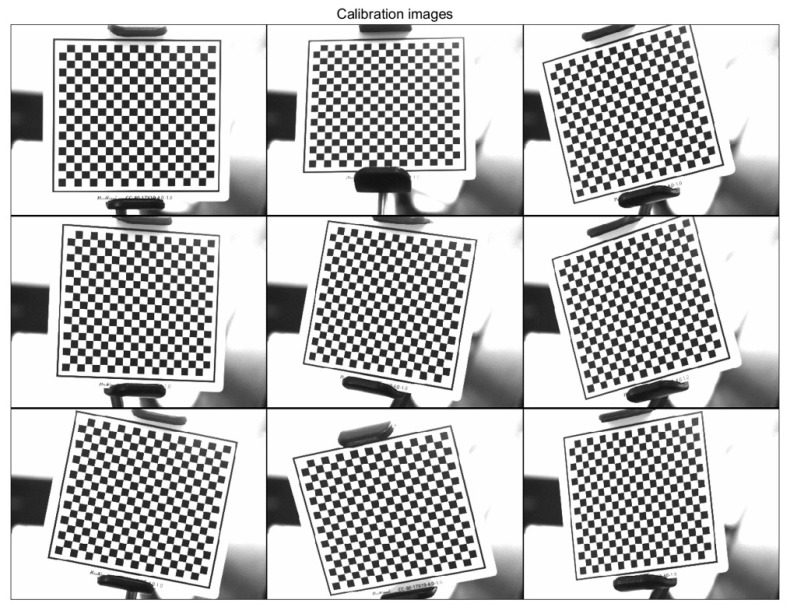
Camera calibration.

**Figure 13 sensors-23-00013-f013:**
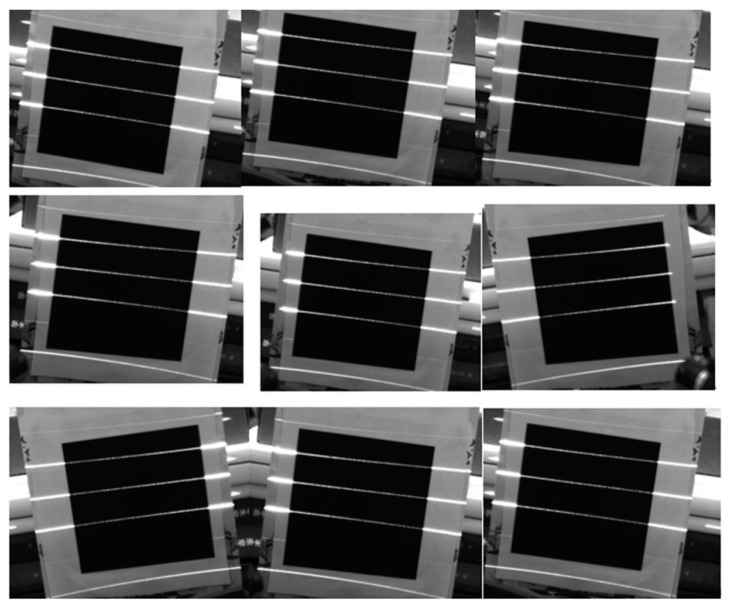
The image of light stripe centerline extraction with black plate.

**Figure 14 sensors-23-00013-f014:**
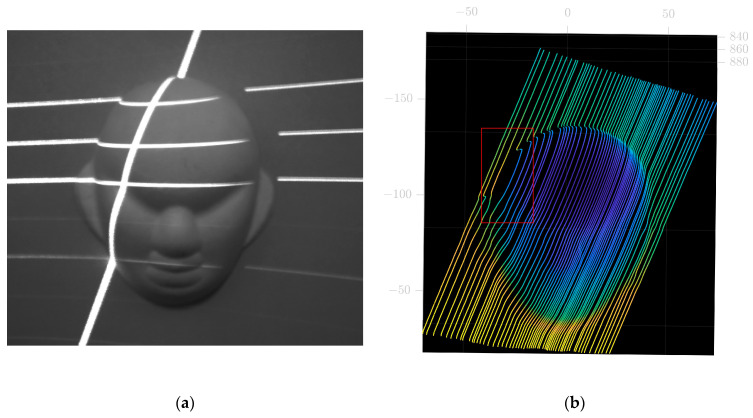
Traditional method of measurement. (**a**) Face mask model from camera view. (**b**) Point cloud image from the traditional method.

**Figure 15 sensors-23-00013-f015:**
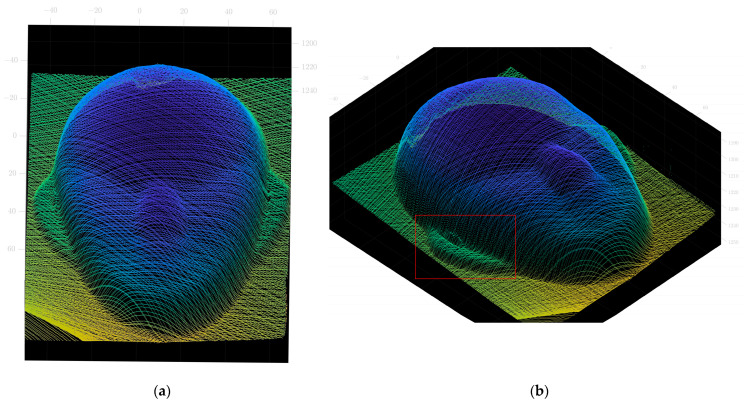
Point cloud image from this paper. (**a**) Point cloud image from frontal perspective (**b**) Point cloud image from the side perspective.

**Figure 16 sensors-23-00013-f016:**
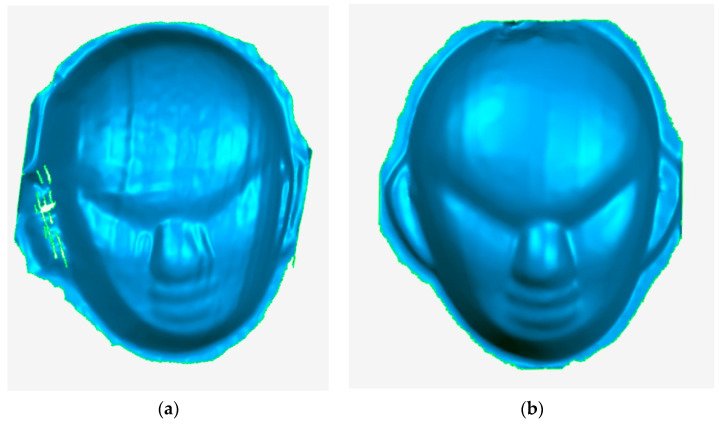
3D reconstruction image. (**a**) Traditional method (**b**) The method of this paper.

**Figure 17 sensors-23-00013-f017:**
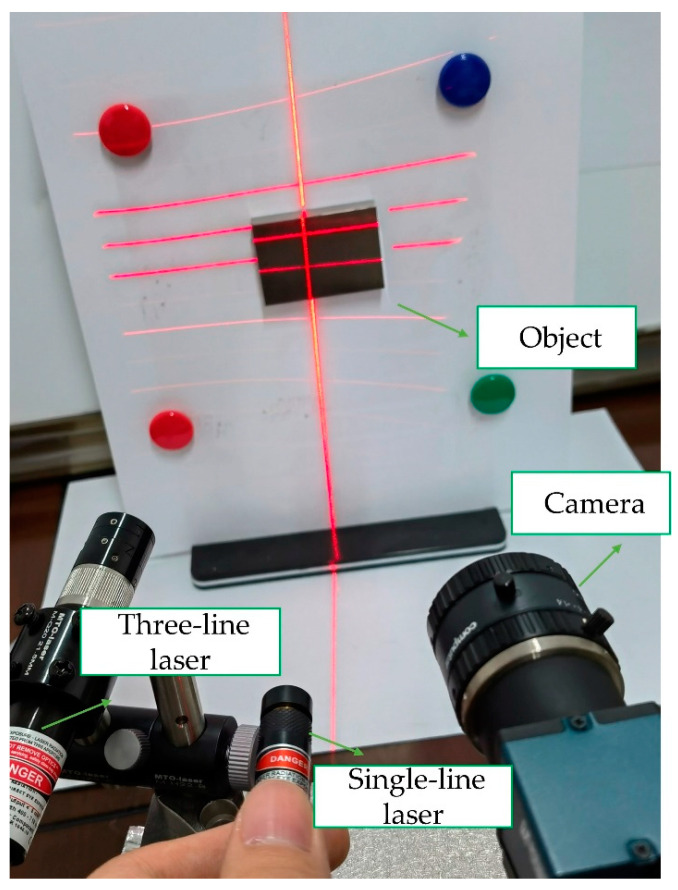
Measurement image of the standard gauge block.

**Figure 18 sensors-23-00013-f018:**
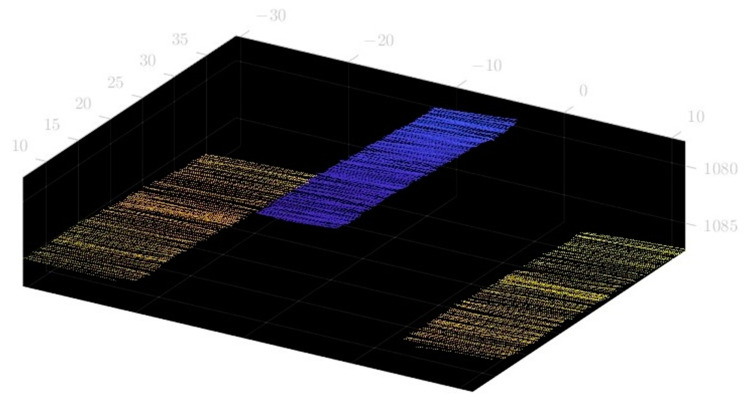
Point cloud image of a gauge block.

**Figure 19 sensors-23-00013-f019:**
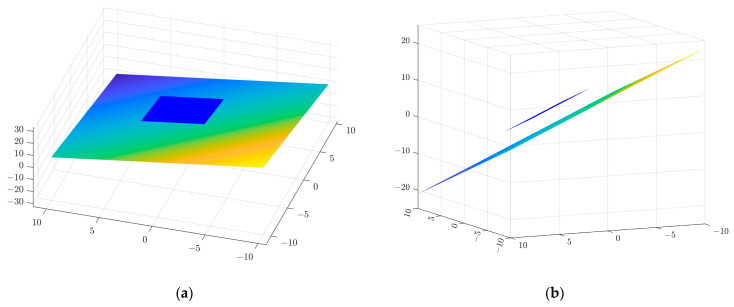
Fitting spatial plane images. (**a**) Top view. (**b**) Side view.

**Table 1 sensors-23-00013-t001:** Measurement results of the standard gauge blocks (mm).

NO.	Thickness			
	70	60	50	40
1	69.920	60.102	50.122	39.902
2	70.101	60.085	50.089	39.893
3	70.063	59.989	49.923	40.113
4	70.042	60.092	49.906	40.136
5	69.929	60.112	50.104	40.104
6	69.918	60.003	50.113	39.915
7	69.892	59.888	49.896	39.911
8	70.102	59.933	50.091	40.163
9	70.097	59.848	50.132	40.072
10	70.114	59.887	49.899	40.085

**Table 2 sensors-23-00013-t002:** Comparison of the measured value with the actual value.

Real	Thickness (mm)			
	70.000	60.000	50.000	40.000
Average	70.025	59.994	50.026	40.030
RMSE	0.0850	0.0957	0.1039	0.1083
STD	0.0856	0.1007	0.1056	0.1099

## Data Availability

The data used to support the findings of this study are available from the first author upon request.

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
