# Peer review of "A Three-Dimensional Structured Light Vision System by Using a Combination of Single-Line and Three-Line Lasers"

_sensors, 2022, doi:10.3390/s23010013_

Round 1
Reviewer 1 Report
A 3D laser scanning strategy combining a single-line and a three-line laser is proposed in this paper. This system supports free scanning of the single-line laser, which has value in non-industrial applications. However, the system framework has significant defects in principle. I cannot consider this work before major revisions and thorough discussions.
(a) In laser scanning system, accurate and reliable 3D reconstruction requires large angle between the baseline and the laser stripe. Therefore, a common layout of the system is that the camera and the laser placed horizontal while the laser stripe projected vertical. However, in the system shown in Fig. 11, the laser stripe is almost horizontal, too. In this case, the ray from the camera almost coplanar with the light plane of the laser. How can you get reliable 3D coordinates in Step 2 of Section 4?
(b) In Step 3 of Section 4, the light plane of the single-line laser is fitted from three intersections, i.e., points a, b and c in Fig. 10. However, in most case, the three intersections are almost collinear, especially in the case of Fig. 11. How do you make reliable plane fitting with three collinear points?
Other comments:
1. The logic of the introduction should be carefully sorted out. The reviews for system & application should be separated with calibration.
2. The standard camera model is well known for most readers, so Section 3.1 could be further simplified.
3. In Eq. (5), why the third plane can be formulated with parameters of the first two planes and the factor μ?
4. In Step 1, when detect the pixel coordinates of the intersection, as shown in Fig. 9, do you consider the case when the surface is not planar?
5. In Fig. 11, the laser stripes of the three-line laser is in the bottom part of the field of view, while the object is in the top part. The field of view is wasted.
6. The advantage in Fig. 16 is trivial and not so convincing.
Author Response
Thank you for your comments, the corresponding changes have been provided in the document.
Please see the attachment.

Reviewer 2 Report
In this paper, the authors proposed a method of three-dimensional measurement system that combines a single-line and a three-line laser. Thus, the proposed method is more convenient and straightforward. In addition, the corresponding qualitative and quantitative measurement results are given in the experiment. However, I think the innovation of this paper cannot meet the requirements of this journals. Finally, give the author some suggestions to make corresponding modifications.
(1)In the experiment, the camera used by the author is MER-040-60 of Daheng, the resolution of this camera is 752 x 480 pixels. However, in the camera calibration results in 267 line , the value of the principal point is 1929 x 1279 pixels, which shown that the calibration result is incorrect or false.
(2)In the experimental part, Figure 11 and Figure 17 are nearly the same. It can be seen in the figure that the main scanned area of the three line laser is a metal cylinder, which is inconsistent with the block gauge. If the block gauge measurement is carried out in the experiment, a separate picture of the block gauge need to be provided, and the measurement 3D data is presented in the paper. In addition, a table of comparing the accuracy of the proposed method with that of the traditional multi line laser method in quantitative analysis is added.
Author Response

(The authors gave the same response as above.)

Reviewer 3 Report
The paper [12] is the same of [20] (row 61).
Row 64: The paper 21 is not written by zeng.
Row 77: “The results of the different parts show that this test works well” – you should explain in a better way the result of the paper you are referring to
Row 81: “In panoramic 3D shape measurement 80 based on a turntable, the point cloud registration accuracy is greatly affected by the 81 calibration of the rotating axis.” This statement is not precise as well, because 3D-laser based scanner technologies are different and not all of them are affected by accuracy of the moving systems to which they are coupled. For example, a structured-light scanner with a turning table is not dependent on the accuracy of the rotary table because there is an alignment algorithm that superimpose every scan to the previous one.
Chapter 3.1: Please, add some information about the literature you refer for equations 2 and 3.
Chapter 3.2: Please, explain the function of the objective function and why the constraint of objective function can ensure a fixed angle between the planes.
figures of the Gauge block point cloud are missing.
Clarify how the thickness measurements were taken
Conclusion
How do you resolve the problem of occlusion and discontinuities? It possible to move the camera during the scan? These arguments should be explained in a more detailed way.
Probably you should also provide dimensional limitations for this structured light scanner. Which is the maximum and minimum dimensional limits that I can acquire whit this set-up?
The work is quite innovative and interesting, but you have to better explain some points of the paper.
Author Response
Thank you for your comments, the corresponding changes have been provided in the document.
Please see the attachment.
Since the website can only upload one word file, we have put the essay and the response together.
The first part is the thesis and the second part is the response.

Reviewer 4 Report
This manuscript proposes a 3D measurement system by using the combination of a still 3 line structure light and a single line with variable scanning direction. The advantage of the proposed approach is being able to deal with occlusions and discontinuities of the target object. The whole work is quite complete and with technical novelty. It is recommended for the publication in Sensors. A minor comment is that the spatial accuracy of 0.16mm can be discussed and analyzed with more detail, rather than just giving a number from the experiment. The mechanism which could influence the spatial accuracy, for example, the line thickness, can be highlighted. This will also give as the guideline for the performance improvement of the proposed method.
Author Response

(The authors gave the same response as above.)

Round 2
Reviewer 1 Report
Although the authors reply all the comments, the issues have not been essentially resolved. The three-line laser takes up a large part of the field of view, and its light plane cannot be reliably estimated. Therefore, only a small object can be scanned, and the quality of the point cloud is not so good.
To establish a 3D laser scanner with the hand-held line laser, the authors may refer to the David Laserscanner (https://en.wikipedia.org/wiki/David_Laserscanner).
Reviewer 2 Report
I don't think the author has made reasonable modifications. In the experiment part, the three line laser should be illuminated on the main measuring object. However, the most central object to be measured in the captured image is a cylinder, which does not meet the requirements of practical application measurement. If the center of the camera filed is the object marked in the figure, the author needs to replace the reasonable experimental image.
Reviewer 3 Report
The paper is acceptable but it would be better to add some of the comments in the cover letter file in the text.
